# Antioxidants in Shell and Nut Yield Components after Ca, Mg and K Preharvest Spraying on Hazelnut Plantations in Southern Chile

**DOI:** 10.3390/plants11243536

**Published:** 2022-12-15

**Authors:** Carlos Manterola-Barroso, Karina Godoy, David Alarcón, Daniela Padilla, Cristian Meriño-Gergichevich

**Affiliations:** 1Scientific and Technological Bioresources Nucleus (BIOREN-UFRO), Universidad de La Frontera, Temuco 4811230, Chile; 2Doctoral Program in Science of Natural Resources, Universidad de La Frontera, Temuco 4811230, Chile; 3Laboratory of Physiology and Plant Nutrition for Fruit Trees, Faculty of Agricultural Sciences and Environment, Universidad de La Frontera, Temuco 4811230, Chile; 4Laboratory of Soil Fertility, Faculty of Agricultural Sciences and Environment, Universidad de La Frontera, Temuco 4811230, Chile; 5Department of Agricultural Production, Faculty of Agricultural Sciences and Environment, Universidad de La Frontera, Temuco 4811230, Chile

**Keywords:** *Corylus avellana* L, hazelnut shell, macronutrients, Tonda di Giffoni, phenolic compounds, waste material

## Abstract

To determine the effects of pre-harvest calcium (Ca), magnesium (Mg) and potassium (K) spraying on the antioxidant activity and capacity of hazelnut (*Corylus avellana* L.) shells, as an approach to sustain the utilization of the main residue derived from this industry, four commercial hazelnut (Tonda di Giffoni) orchards located in Southern Chile (Cunco, Gorbea, Perquenco and Radal), during the 2018/19 season were sprayed three times with five combinations of Ca (300 and 600 mg L^−1^), Mg (300 and 600 mg L^−1^) and K (300 and 600 mg L^−1^). Yield components were determined in harvested whole nuts, whereas Ca, Mg and K concentrations, as well as total phenolic compounds, free radical scavenging antioxidant activity, and oxygen radical absorbance capacity, were determined in shells. All spray treatments with both Ca, Mg and K combinations showed a significant interaction between locality and treatment (L × T) on increased stabilized nut yield (kg ha^−1^) in comparison with the control treatments, whereas nut quality was differentially affected by treatment and orchard locality, thus confirming a significant L × T relationship regarding nut length and kernel yield. However, locality showed a major effect on kernels and shells. A significant relationship was determined between locality and the Ca, Mg and K spraying (L × T) concerning antioxidant compounds such as phenolics, whose amounts exceeded those of the control treatments by three times. Antioxidant capacity and activity in shells showed a significant L × T relationship, and nutshells collected from Cunco showed remarkably (3–4 times) higher levels of these factors than the other evaluated localities. Interactions between spray treatment and orchard location were responsible for the different values obtained in the experiments, confirming the influence of the environment on the efficacy of the treatments. Finally, these shells are worth further study as an interesting residue of the hazelnut industry due to their nutritional and antioxidant properties.

## 1. Introduction

The annual world production of hazelnuts (*Corylus avellana* L.) exceeds one million tons (t), and Chile has become the main producer in the Southern Hemisphere, with an off-season nut supply of approximately 32,949 t in 2021 [1]. Chilean plantations cover about 30,000 hectares (ha) of land, mainly concentrated in southern regions such as El Maule and La Araucanía; market projections point to 60,000 ha of land by 2030, with an estimated production of 127,000 t year^−1^ [1,2,3,4,5]. In fact, the hazelnut seed or kernel has gained attention due to its unique taste and its contents of bioactive and health-promoting compounds, such as phenolic acids, essential amino acids, unsaturated fatty acids, and calcium (Ca^2+^), magnesium (Mg^2+^), and potassium (K^+^) [6,7,8,9,10,11]. Indeed, 90% of hazelnuts produced in Chile are marketed without shells (unshelled) [1], though their hard shells have a mass value of over 50% of the total weight of stabilized nuts and are becoming the main hazelnut production by-product, which means that about 400,000 t of shells are produced worldwide ever year [5,11,12,13,14]. Currently, this biomass is mainly used as an energy source for heaters due to its low cost [15]. However, due to continuous efforts for the upcycled reutilization of potential renewable materials in agriculture, this waste material could be valorized as a promising opportunity in the context of a circular bio-economy [14,16]. This waste shell exhibits significant hardness and toughness, and it is structurally composed of hemicellulose (25–30%), cellulose (26–32%), lignin (40–43%), and extractives (3.3–4%) [14,15,17], which makes it a potential and interesting raw material for innovation and a reliable source of natural antioxidant compounds [7,18]. Researchers have explored the utilization of hazelnut shells for the extraction of phenolic compounds in order to obtain natural value-additives [13,15,19]. Accordingly, Contini et al. [7] reported a phenolic compound content of 2.7 mg g^−1^ DW in shells for the hazelnut cultivars of Tonda Gentile Romana, Tonda di Giffoni (TDG), Tonda Gentile delle Langhe and Tombul processed at different roasting temperatures and times. However, this research did not discriminate between localities and how orchards were managed. Stévigny et al. [12] performed an optimum extraction method and reported a similar content of phenolic compounds (3.00 ± 0.21 GAE mg g^−1^ of shell) detected in TDG shell samples. Although this material has shown richness in phenolics, little is known about the other present antioxidants; therefore, it is worth studying how the chemical and biochemical composition of shells is influenced by the environment and the crop management of hazelnut trees grown in commercial plantations.

The planting of this nut crop in new plantation areas might affect its productivity, leading to reductions in the nutrient uptake capacity required for adequate vegetative growth and the productive yields of trees [20,21]. In Southern Chile, agricultural activities are mainly conducted on acidic Ultisols and Andisols (pH ≤ 5.5) [22,23,24], associated with a low natural availability of cations such as Ca^2+^, Mg^2+^ and K^+^ [25,26]. Against this background, hazelnut grower efforts are aimed at ensuring enhanced fruit sets, cluster retention, kernel yields, and industrial nut traits [12,27]. Accordingly, the foliar spraying of these macronutrients has become routine, with several administrations during the growth season as part of the annual nutrition programs of commercial orchards [21,28]; such treatments have demonstrated effectiveness at low rates, with a uniformity of application and a very quick plant response [29,30,31]. In this context, these macronutrients are involved in several metabolic functions [27,32,33,34,35]. Calcium may be involved in plant tolerance to environmental stresses via the regulation of antioxidant metabolism [24,36]. Cross-talk between Ca^2+^ and reactive oxygen species (ROS) leads to effective antioxidant defense and the enhancement of antioxidant compound levels [16,37]. It has also been shown that this crop requires soils with relatively high amounts of extractable Mg, which shows positive effects on long-term tree health, vigor, and immediate yield [38]. Özenç and Özenç [30] reported that the fertilization of the Tombul hazelnut tree cultivar with MgSO_4_·7H_2_O (150 to 250 kg ha^−1^) planted in the Black Sea region showed a lesser loss of kernel quality and increased total production, particularly at the highest rate (250 kg ha^−1^), compared with controls. The Mg fertilization of hazelnuts must be carefully conducted to prevent the loss of kernel features [30]. These authors also stated that the kernel yield (%) of the Tombul hazelnut cultivar was reduced by 3.4% at the highest Mg rate. Meriño-Gergichevich et al. [21] reported a K value of 0.6% in the kernels of the TDG hazelnut cultivar planted in Southern Chile, and K was found to be strongly correlated with Mg content in the kernel, as well as with nut and kernel weights.

There are several nutrient application programs for hazelnuts, but there is no information about the features of shells, which act as the natural protection of kernels from environmental, phytopathological and mechanical constraints [4,39]. Hence, the main objective of this work was to study the effects of a complementary nutrition program on the industrial yield and nutshell antioxidant features of four hazelnut plantations. A further aim was to understand the behavior of the nutshells and how they are affected by orchard locality and plant nutrition program.

## 2. Results

### 2.1. Stabilized Nut Yield

The nut yield per hectare (kg ha^−1^) and plant (kg pl^−1^) of whole nuts (shelled) stabilized at 6% humidity and harvested from plants grown in the four studied orchards are shown in Figure 1A and 1B, respectively. An interaction was found between locality and all spray treatments (L × T), demonstrating a significant increase in yield per ha in all commercial orchards located in Cunco, Gorbea, Perquenco, and Radal in comparison with the control plants (Figure 1A). In terms of nut yield, Gorbea showed the highest rates, exceeding 2000 kg ha^−1^ under the 300+K, 600+K, 300, and 600 treatments compared with the control plants (*p* ≤ 0.05), as well as showing an average increased yield of 37.5% compared with the other sprayed plants. Similarly, Perquenco showed a significant 32% increase in nut yield (kg ha^−1^) after all spray treatments. Furthermore, Radal showed a 30% higher nut yield following the spraying of Ca and Mg at 300 and 600 mg L^−1^ (300 and 600, respectively) than that obtained from control plants (*p* ≤ 0.05). However, the addition of K (300+K and 600+K) to the Ca and Mg treatments resulted in 14% and 22% yield increases, respectively, in comparison with the control plants (*p* ≤ 0.05). Figure 1B shows the stabilized nut yield per plant (kg pl^−1^) of plants harvested three months after spray treatments, with Cunco and Gorbea (571 plants ha^−1^) having higher yields than Perquenco (667 plants ha^−1^) and Radal (500 plants ha^−1^) (*p* ≤ 0.05). The application of Ca and Mg at both rates demonstrated a significantly better nut production rate compared with the control plants (*p* ≤ 0.05), whereas the incorporation of K (300+K and 600+K) did not lead to any changes in this variable for the four studied localities.

### 2.2. Nut, Shell and Kernel Traits

Table 1 describes the commercial traits of the harvested hazelnuts (nut, shell and kernel) from the studied localities after the spraying of Ca, Mg and K. Industrial components such as nut weight (g), length (mm), width (mm), shell weight (g) and thickness (mm), and kernel yield (%) were measured in stabilized nuts. Regarding nut weight, under all nut treatments, Cunco and Gorbea showed means of 2.71 g (2.38–3.17 g) and 2.66 g (2.48–2.69 g), respectively, representing 10.4 and 8.5% higher nut weights than Perquenco and Radal, respectively (*p* ≤ 0.01). Plants sprayed with 300+K showed better nut weights compared with the control plants (*p* ≤ 0.05). For nut length, a strong interaction was determined between locality and spray treatment (L × T), with Gorbea showing 1.72, 7.4, and 7.29% taller nuts than Cunco, Perquenco, and Radal, respectively (*p* ≤ 0.001). Moreover, the nut lengths of nuts harvested from the Cunco and Gorbea orchards were significantly increased following the 300+K and 300 treatments (*p* ≤ 0.05). Concerning nut width, Gorbea showed means that were 3.7% higher than those of nuts collected from the other localities (*p* ≤ 0.01). The width/length ratio of the harvested nuts from Perquenco and Radal registered a roundness index (RI) of 0.91, which represented a globular nut shape (data not shown). In terms of morphological description, the average kernel height was 18.8 mm, though the values in Cunco and Gorbea were higher.

The shell weight determined in Cunco was 11.1, 20.5, and 10.6% higher than in Gorbea, Perquenco, and Radal, respectively (*p* ≤ 0.001), although no treatments were able to modify this variable in harvested nuts. The highest thickness of shells was found in Gorbea (1.82 ± 0.05), which was 0.19 mm more than that measured in Perquenco (1.63 ± 0.04). Likewise, as for the shell weight, we did not find any influence of the spray treatments.

The kernel/shell ratio percentage (42%) was found to strongly favor the shell residue material (without discounting the weight of gases and other interferences that may have minimally affected the ratio), reaching an approximate average of 59% of stabilized biomass relative to the total weight of whole nuts. Finally, it was noted that locality moderately and strongly affected all analyzed parameters. However, the treatment variable only resulted in a significant effect for the kernel height (Table 1).

### 2.3. Calcium, Mg and K Concentration in Shell Samples

Figure 2 shows the Ca, Mg and K concentrations (mg 100 g^−1^ DW) of the hazelnut shells obtained from the evaluated orchards in La Araucanía. A significant interaction (L × T) was found regarding the Ca concentration in shells harvested from Perquenco in comparison with other orchards, as the 300 and 600 treatments resulted in 13.5 and 19.6% higher contents, respectively, in comparison with 600+K (*p* ≤ 00.01; Figure 2A). Gorbea and Radal showed similar Ca contents in shells, but no differences were observed in the concentration of this macronutrient in the shells of nuts collected from Radal plants. In Cunco, the harvested nuts showed significant increases in Ca under the 600+K and 300 treatments compared with the control treatment (*p* ≤ 00.05). Furthermore, the K and Ca concentrations showed a significant interaction between both factors (L × T) that was even stronger than that for Ca contents. These results prompted the further investigation of the importance of orchard locality over other parameters. In the Cunco orchard, we noted significant increases in the K and Ca contents of shells after the 600+K and 300 treatments (600 mg L^−1^ of Ca, Mg, and K and 300 mg L^−1^ of Ca and Mg, respectively), with increases of 45% and 200% in total K concentration for the 600+K and 300 treatments, respectively, in comparison with the control plants (*p* ≤ 0.05). On the other hand, the total content of Ca increased by approximately 25% and 30% following the 600+K and 300 treatments, respectively, compared with the control treatment. In this case, a strong interaction between the factors of locality and treatment (L × T) was determined, so it was possible to confirm an effect of the treatments on the elemental composition of the shells (particularly regarding Ca and K).

In Gorbea, there were no significant differences in the K and Mg contents (%) of the spray treatments compared with the control treatment. However, in both Gorbea and Cunco, we observed significant increases in Ca concentration, though at low values (<10%), following all evaluated spray treatments compared with the control treatment (*p* ≤ 0.05). Perquenco showed a very interesting result in comparison with the other evaluated locations; significant decreases in Ca and K concentrations were observed, particularly in the case of the 600+K treatment in comparison with the control treatment. Moreover, the effect produced by the 600 treatment was striking, showing a significant increase in the rate of Ca in comparison with the control plants.

### 2.4. Total Phenolic Compounds

Regarding the contents of the total soluble phenolic compounds (TPC) in hazelnut shells (Figure 3), an important relationship was determined between locality and the Ca, Mg and K treatments (L × T). In addition, the evaluated treatments and localities separately demonstrated significance (*p* ≤ 0.05). In the case of the orchard located in Cunco, we determined phenolic concentrations of up to three times higher than those of the other evaluated orchards (for all evaluated treatments). Perquenco and Radal showed increased total concentrations of soluble phenolic compounds for the first evaluated treatment (Ca, Mg and K 300 at mg L^−1^) in comparison with the control treatment. Finally, the orchard in Gorbea only showed differences between the 300 treatment (Ca and Mg 300 at mg L^−1^) and the control treatment (*p* ≤ 0.05).

### 2.5. Free Radical Scavenging Activity

For the radical scavenging activity (RSA) in hazelnut shells (Figure 4), a significant interaction was determined between locality and spray treatment (L × T). The RSA was significantly higher in samples harvested from the Cunco orchard than the other localities (*p* ≤ 0.05). However, there were no significant differences in relation to the evaluated treatments. In Perquenco, we determined a negative effect of the 300+K and 600+K treatments on the RSA in comparison with the control treatment (*p* ≤ 0.05), whereas the effect of the 300 and 600 treatments on RSA was not noticeable in respect to the control treatment. In Radal, the opposite behavior was observed, as the RSA of shells increased by 55%, 53.3%, 53%, and 74.6% following the 300, 300+K, 600+K, and 600 treatments, respectively, compared with the control treatment. Finally, in Gorbea, significant differences in the shell RSA were only observed between plants sprayed with the 600+K treatment and plants in the control treatment, with an increase of 2%.

### 2.6. Antioxidant Capacity:

For antioxidant capacity (AC), an interaction between the treatment and locality factors (L × T) was found (*p* ≤ 0.001). Cunco showed the highest AC in shells (Figure 5). In this orchard, trees sprayed with the 300+K treatment showed similar AC values as the control plants, but other treatments showed lower AC levels, e.g., the 600 treatment showed a reduction of 56.4% (*p* ≤ 0.05). Gorbea and Perquenco also showed similar behaviors, with 36% and 53% reductions in AC compared with the control treatment (*p* ≤ 0.05). However, the application of the 300 treatment led to the same AC level as the control treatment in both above-mentioned hazelnut orchards. There were no statistical differences in the determined AC of control treatment shells in the Gorbea, Perquenco and Radal orchards. However, Perquenco showed a substantial decrease in AC for the 600 treatment plants compared with those in the other localities (*p* ≤ 0.05), mainly Cunco (*p* ≤ 0.05). Only in the Radal orchard was the shell AC considerably increased following the 300+K (22.5%), 300 (29%), and 600 (43.4%) treatments in comparison with the control shells (*p* ≤ 0.05).

### 2.7. Relationship of Mineral Content and Shell Traits

To understand the effect of sprayed Ca, Mg and K on antioxidant properties in our study, a Pearson’s correlation test was conducted for the data on Ca, Mg and K concentrations in shells and the determined TPC, RSA and AC values (Table 2). Strong relationships were found between Ca and K in hazelnut shells at the Cunco (r = 0.79), Perquenco (r = 0.75), and Radal (r = 0.75) orchards, whereas Mg was only correlated with Ca (r = 0.93) and K (r = 0.71) in shell samples collected from Radal. Both Ca and K concentrations in shells were negatively correlated with shell weight in Cunco and Radal, whereas Mg showed a significant relationship with this evaluated parameter (r = 0.54) in Cunco and with shell thickness (r = 0.57) in Gorbea. No relationships were established between mineral concentration and kernel yield, TPC, RSA, or AC.

### 2.8. Response of Antioxidant in Shells

Figure 6 shows that the effects of shell TPC and RSA on AC were different in each locality. In Cunco, high shell TPC values were associated with high RSA and AC values, and in Radal, TPC was associated with AC but not RSA. In those orchards, shells with TPC values of over 270 µg GAE g^−1^ DW led to higher antioxidant activity and capacity values, as measured with the DPPH and ORAC methodologies. Perquenco, which had lower levels of TPC, also showed increased RSA and AC values, although no strong correlation of shell TPC and RSA with AC was observed. On the other hand, a strong relationship between TPC and RSA was determined for Cunco (r = 0.99), Gorbea (r = 0.46), and Radal (r = 0.51). Furthermore, shell TPC was only correlated with AC in shells from nuts harvested in Cunco.

## 3. Discussion

### 3.1. Yield and Industrial Component of Nuts

Hazelnut is not a high-yield crop, with an expected potential yield of 4.5 t ha^−1^ [33]. For orchards planted in Mediterranean and temperate areas such as El Maule and La Araucanía (regions that lead in the number hazelnut plantations), the mean hazelnut production was found to be 1.24 t ha^−1^ from 2016 to 2021 [1,21]. The total 2021 production of the TDG hazelnut cultivar in Chile was 17,371 t or 52.7% of annual Chilean production [21]. In our experiment, all orchards showed average yields of 1.20 t ha^−1^ for control plants, and the spray treatments increased nut yields in Cunco, Gorbea, Perquenco, and Radal by 28%. The 600+K and 600 treatments, in particular, showed 30% and 31.6% more stabilized nut yields, respectively, in comparison with the control plants (Figure 1A). Similarly, control plants averaged 2.13 kg ha^−1^ of observed yield per plant (kg pl^−1^), which was 28% less than the 300+K (2.75 kg pl^−1^), 600+K (3.04 kg pl^−1^), 300 (2.95 kg pl^−1^), and 600 (3.11 kg pl^−1^) treatments. Plantation locality showed an effect on nut yield, with the orchard in Perquenco (5 × 3 m) exhibiting a yield of 1.12 kg pl^−1^ for the control plants, which was 52% lower than the yields of the Cunco (2.91 kg pl^−1^), Gorbea (2.42 kg pl^−1^), and Radal (2.09 kg pl^−1^) (Figure 1B) hazelnut orchards. The main industrial yield component was associated with kernel yield (kernel/shell ratio) in the TDG cultivar previously studied by Meriño-Gergichevich et al. [21], who reported an average kernel yield of 42%, which was similar to the yields reported in this study for Gorbea and Perquenco (Table 1). These results demonstrated significant effects of plantation locality on nut weight, shell weight, and shell thickness. Regarding shell properties, thickness and weight showed the lowest values in nuts harvested from the Perquenco orchard. Moreover, Ca and Mg showed a differential relationship concerning shell thickness (r = 0.57 and r = −0.50, respectively) for the Gorbea and Perquenco orchards, indicating that this variable was not affected by the Ca, Mg or K concentrations in the shells (Table 5). In Cunco, Mg concentration showed a positive relationship (r = 0.54) with shell weight and Ca concentration showed a negatively relation with shell weight (r = −0.63).

### 3.2. Ca, Mg and K Concentrations in Shell Samples

A previous study on hazelnuts reported leaf and kernel concentrations of Ca, Mg and K for trees grown in Chile and Turkey [21]. Nevertheless, information about the mineral composition of shells has rarely been reported. Our experiment showed a significant L × T interaction regarding Ca in shells, as higher amounts of Ca were found in nuts from Perquenco for plants following the 600 mg L^−1^ treatment, and Gorbea showed a similar behavior. In Cunco, Ca was increased following the 600+K and 300 treatments. In untreated samples, the Ca content in kernels was equivalent to 25% of that reported in leaves. Moreover, the Mg contents in kernels were similar to those obtained in leaf samples. Furthermore, plantation locality was found to be determining factor of kernel thickness and width, as well as the shell/kernel ratio, probably due to the contributions of soil and climatic conditions on the synthesis process of the pericarp (Table 1).

Finally, the K contents in kernels were 50% of the values obtained for leaves.

### 3.3. Total Phenolic Compounds

Gorbea showed a decreasing TPC trend after the first treatment (300+K), with values of 8% less than those of the control treatment (Figure 3). Our results correspond with the results reported by Tekaya [40] for European olive (*Olea europaea)* cv. Picholine, who obtained a TPC reduction of 26.2% in comparison with the control plants after a spraying treatment (50 g L^−1^ of Mg; 3 L ha^−1^). The differences found in the Gorbea orchard following the 300+K treatment in our study may be attributed to the addition of K in this particular case, as also demonstrated by Nguyen et al. (2010) [41], who postulated an increase in TPC (mg GAE g^−1^ DW) in basil leaves (*Ocimum basilicum* L.) corresponding to three cultivars (Sweet Thai, Dark Opal, and Genovese) following different K fertilization treatments (1–5 mM); those authors found an increasing trend that was directly proportional to the K concentration, and they highlighted a strongly significant increase of 27.6% in TPC (*p* ≤ 0.05) compared with the control treatment in the case of the cv. Dark Opal variety after a 5 mM K treatment. 

Contrary to what was stated by the above-mentioned authors [40,41], Perquenco and Radal showed slightly increased trend in the total phenolic concentration following the first evaluated treatment (300+K) compared with the control treatment (12% and 11%, respectively) in our study. Nevertheless, it is impossible to have total confidence in results obtained during the first or second year or season of growth. It is necessary to study at least three or four years of treatment seasons to obtain solid data, as well as to follow up during after the total treatment trial time each season and year.

### 3.4. Free Radical Scavenging Activity

To determine antioxidant activity, we used the DPPH free radical scavenging methodology, mainly due to its widespread use in the characterization of the antioxidant activity of vegetable materials and extracts [42,43]. In this study, this methodology did not reveal any strong correlations regarding the AC of hazelnut shell samples, as shown in Figure 6. For this reason, it was important to use other descriptive methodologies for antioxidant property evaluations. For instance, we also used the quantification of total phenolic compounds methodology and demonstrated a strong interaction between the AC of hazelnut shell samples and locality, mainly in Cunco and Gorbea, probably due to physiological responses against the abiotic conditions of the environments of the evaluated commercial orchards (Figure 6). Our results regarding the effect of Ca on antioxidant activity coincided with the results obtained by Reyes et al. [44], who found an increase in the RSA of blueberry, cv. Bluegold, leaf samples following calcium sulfate (CaSO_4_) application at rates of 5 and 2.5 mM, with the latter showing a significant increase in RSA after 15 days of experiments.

Regarding hazelnut kernels (edible part), Meriño-Gergichevich et al. [21] reported RSA kernel concentrations of around 5 mg TE g^−1^ FW in their control treatments of the Tonda di Giffoni (TDG) cultivar. However, in this study, the control treatments at Cunco, Perquenco, and Gorbea tended to show dramatically decreased RSA kernel concentrations (around 1.5, 0.75, and 0.5 mg TE g^−1^ DW, respectively), although not too much in the case of Cunco. 

### 3.5. Antioxidant Capacity

To determine the AC of the hazelnut shell extracts, we used the ORAC (Oxygen Radical Absorbance Capacity) methodology, which is categorized as a hydrogen proton transfer methodology (HAT) based on the stabilization of a free radical (AAPH or 2,2′ azobis-(2-amidinopropane)) through the action of an antioxidant and the complete transfer of a hydrogen proton. It directly depends on the damage of a free radical (AAPH) to a fluorescent probe (sodium salt fluorescein), which produces a degenerative change in the intensity of fluorescence [45]. As there are no published results related to this methodology on hazelnut shell extracts, the results obtained for our evaluated shells were compared with values of seeds and kernels. The data obtained in related projects [46,47] showed an average shell AC value of 50 µmol TE g^−1^ DW, which was extremely low in comparison with the average shell values reported in this study (control treatment AC mean: 2.119,7 µmol TE g^−1^ DW), which were 42 times than the AC values of the kernel.

Regarding the determination of the total antioxidant activity and capacity of our hazelnut shell samples, we were able to establish significant differences between the applied spray treatments in each location. We developed three metrics of biochemical character—total antioxidant activity, total phenolic compounds, and total antioxidant capacity—to determine a way in which the antioxidant potential present in hazelnut shells and the effect of nutrition management could be oriented to enhance their antioxidant properties. In addition, we concluded that the locality of the plantation has a very relevant role in the resultant antioxidant properties of hazelnut shells.

### 3.6. Relationships

In this study, it was possible to determine a strong relationship between the concentration of the total phenolic compounds and AC of hazelnut shells, though this may depend on the phenolic qualities of shells and could be explained by the behavior of the shells in the studied orchards in response to local environmental conditions. We also established a potential physiological effect on nut rigidity and firmness, thickness, yield, and other shell traits (Table 1). We noted that the plantation locality moderately and strongly affected the commercial properties of the nuts. However, this was not the case for the spray treatment, which only resulted in a significant effect on kernel height. Additionally, we determined a very interesting effect of locality regarding AC, TPC, and RSA, with lesser values at orchards with lower frequencies of agronomic and nutritional management (Cunco and Gorbea). Therefore, we can definitively conclude that the plantation locality is more determinant than agronomic management on the antioxidant properties and nut shell characteristics of hazelnuts.

Figure 6 illustrates the relationship between RSA and AC, and it is imperative to note that the shells collected from the four evaluated orchards showed completely different behaviors in response to the evaluated spray treatments. These results evidence the need to investigate the importance of plantation locality and environmental conditions in the production and nut traits of hazelnut plants.

## 4. Materials and Methods

### 4.1. Spray Treatments and Fruit Harvest 

Our experiments were performed in commercial hazelnut plantations of four localities of La Araucanía: Cunco (38°58′19.8″ S, 72°07′33.0″ W; altitude: 400 m.a.s.l), Gorbea (39°04′.6″ S, 72°41′48.7″ W; altitude: 115 m.a.s.l), Perquenco (38°25′.2″ S, 72°21′.9″ W; altitude: 299 m.a.s.l), and Radal (39°01′06″ S, 72°19′11″ W; altitude: 168 m.a.s.l). Plants of the Tonda di Giffoni cultivar (eight to ten years old) in a multi-stem training system were planted in regular rows in frames of 5 × 3.5 m (571 plants ha^−1^) in Cunco and Gorbea, frames of 5 × 3 m (667 plants ha^−1^) in Perquenco, and frames of 5 × 4 m (500 plants ha^−1^) in Radal. During the 2018/19 season, all plantations were sprayed three times with five combinations of Ca (300 and 600 mg L^−1^), Mg (300 and 600 mg L^−1^), and K (300 and 600 mg L^−1^), as detailed in Table 3. The treatments were applied at a frequency of 15 days in the phenological growth/premature stage (December 2018 to January 2019) using a backpack spraying machine (Nubola 1200, Cifarelli S.p.A, Voghera, Italy) with a capacity of 17 L and an output of 5 L min^−1^. The water consumption ranged between 1200 L and 1500 L ha^−1^, as determined by canopy volume (4πab^2^, a: ½ height; b: ½ width). Ca, Mg and K were sprayed using commercial Organichem^®^Ca, Organichem^®^Mg, and Organichem^®^K, respectively (Chemie^®^ S.A., Providencia, Chile), which are certified for organic production (Table 3). The Ca, Mg and K concentrations in the water used for spraying are shown in Table 4 (based on standard methods for the examination of water and wastewater (APHA.AWWA.WEF. 22nd edition 2012)). Spraying was performed on clear days without wind or precipitation 72 h before and after harvest in the studied localities.

In each studied orchard, nut harvest was performed in late March and the beginning of April (2019) with two backpacks (Cifarelli S.p.A V1200, Voghera, Italy). Then, the collected nuts were gently transported to the Laboratory of Plant Physiology and Nutrition in Fruits Crops (BIOREN-UFRO), washed with deionized water, and prepared for the stabilization process at 40 °C in forced-air ovens (Heratherm OGS100, Thermo Scientific, Waltham, MA, USA; Memmert UF-55 Büchenbach, Germany) until reaching 6% humidity. Then, they were stored at room temperature until analysis.

### 4.2. Yield Component Analyses

All harvested and stabilized in-shell nuts (per treatment and replication) were weighed to determine the yield per hectare (kg ha^−1^) and plant (kg pl^−1^). One hundred and fifty in-shell nuts randomly sampled from each replication were selected to measure the nut and kernel length (mm), width (mm), and thickness (mm) using a digital caliper (CALDI-6MP, Truper, Mexico). Then, the nuts were cracked by hand, and shell thickness (mm) was measured on the convex side of each half. The in-shell nuts, kernels, and shells were weighed on a precision balance (Snug III-3000, Jadever Weightech, Inc., Vaughan, ON, Canada). All hazelnut nut descriptors were evaluated following the 2008 Bioversity International and FAO guidelines.

### 4.3. Chemical Analyses

Soil samples were collected from 0 to 30 cm in depth and analyzed before the start of the experiments (30 September 2018) (Table 5). The soil pH, Ca, Mg and K were analyzed according to the protocol of Sadzawka et al. [48].

The nutrient concentrations of the stabilized nuts were determined with the method of Sadzawka et al. [48]. Nuts were dried in a forced-air oven (Memmert model 410, Schwabach, Germany) for 96 h at 40 °C until a constant dry weight (DW) was reached. Samples were weighed and burned to ash for 8 h at 500 °C (JSMF-30 T, electric Muffle Furnace of JSR Research Inc., Gongju, Korea). Later, the ashes were digested with 2 M hydrochloric acid and filtered. All samples were measured with a simultaneous multi-element atomic absorption spectrophotometer (model UNICAM 969 Atomic absorption Spectrometer, Ilminster, UK).

### 4.4. Shell Sample Processing

Shells were subjected to a mechanical peeling process using a traditional nutcracker, and then each sample was milled and sieved (<1 mm) in an ultra-centrifugal mill (Retsch ZM 200, Haan, Germany) to obtain shell powder. Next, each sample of milled shell (5 g) was suspended in 20 mL of MeOH (70% *v*/*v*), and the solution was sonicated (Elma S10H, Hohentwiel, Germany) for 30 min with iced water (4 °C) at 37 Khz. Later, the samples were centrifuged (Eppendorf, 5804, Hamburg, Germany) at 8.000× *g* (4 °C) for 10 min and then shaken at 200 rpm (15 °C) in darkness for 18 h (Orbital Shaker ZHWY-100B Zhicheng, Shanghai, China). After this procedure, the solution was re-centrifuged at 8.000× *g* for 10 min. Finally, the samples were passed through a filter with a pore size of 0.45 µm (BIOFIL Syringe Driven, Guangzhou, China) and stored at −20 °C until analysis. Each pellet was resuspended for re-extraction under identical conditions (only 6 h), resulting in a total extract volume of 40 mL.

### 4.5. Total Phenolic Compounds

The total phenolic compounds (TPC) were quantified with the colorimetric Folin–Ciocalteu methodology described by Singleton and Rossi [49] with minor modifications. Based on preliminary laboratory results, we decided to use Na_2_CO_3_ (20% *w*/*v*) (Merck KGaA, Darmstadt, Germany). Analyses were performed with a UV/VIS spectrophotometer at a wavelength of 765 nm (SP-8001, Metertech, Taipei, Taiwan), and absorbance data were analyzed and interpreted via interpolation into a gallic acid (Merck KGaA, Darmstadt, Germany) calibration curve (0 to 250 µg mL^−1^).

### 4.6. Free Radical Scavenging Activity

To determine the radical scavenging antioxidant activity (RSA) of the samples, the DPPH (2,2-diphenyl-1-picrylhydrazyl) methodology proposed by Brand-Williams [43] was adapted for transparent microplates (Jet BIOFIL, Guangzhou, China) in a multi-mode reader (Synergy H1 Hybrid, Winooski, VT, USA) at 517 nm. Additionally, a calibration curve was developed for Trolox (Sigma-Aldrich, Dorset, UK) (0 to 400 µM), and the DPPH reagent (Sigma-Aldrich, Dorset, UK) was diluted from a stock solution (81 µM) at 1:3 *v*/*v* or to an absorbance of 0.9–1.0. All data are expressed as µmol Trolox equivalents (TE) g^−1^ DW. 

### 4.7. Oxygen Radical Absorbance Capacity (ORAC)

To determine the antioxidant capacity of the studied hazelnut shells, we used the “Methodology for the determination of ORAC antioxidant capacity in the pericarp of European hazelnut fruits”, owned by the Universidad de La Frontera and registered in the Chilean Department of Intellectual Rights (DDI), N° 2021-A-8614. This procedure was performed with a multi-mode reader (Synergy H1 Hybrid, Biotek, Winooski, VT, USA) in tetraplicate. We began by preparing working solutions: 2,2′-azobis(2-amidinopropane) dihydrochloride (AAPH) (153 mM) (Calbiochem, San Diego, CA, USA) and 70 nM fluorescein (Sigma-Aldrich, St. Louis, MO, USA), which was obtained from a stock solution at 393 µM. After preliminary tests, it was evident that the extract had to be diluted (1:50), so a Trolox calibration curve (0 to 300 µM) was used. All data are expressed as µmol Trolox equivalents (TE) g^−1^ DW

The methodology was conducted with Gen5™ software. First, we conditioned the equipment to 37 °C. Then, we used injector 1 to immediately dispense 150 μL of 70 nM fluorescein, followed by orbital shaking for 10 s at the maximum intensity and incubation for 15 min. Next, we injected 25 μL of AAPH with injector 2, followed by orbital shaking for 50 s at the maximum intensity, and we finally conducted kinetic readings every 60 s for a total of 2.5 h. Simultaneously, 25 μL of each sample was loaded into standard or blank, 96-well black TPC microplates; we filled the external wells with 250 μL of distilled H_2_O and then inserted the loaded plate into a multi-plate reader at 37 °C. Finally, the data were reduced and interpreted by two equations to calculate the area under the curve (AUC).

### 4.8. Experimental Design and Statistical Analysis

The experimental design was a completely randomized factorial scheme where the factors were treatment and location (5 × 4 × 3). The values obtained in the determination of RSA, TPC, and AC (5 × 4 × 4 factorial model for TPC and RSA and 5 × 4 × 3 factorial model for AC) were subjected to a normality test and a two-way analysis of variance (ANOVA), and a comparison of means according to the Tukey post-hoc test showed a significance level of *p* ≤ 0.05. Finally, a Pearson’s correlation test was performed to determine the linear relationship between the analyzed dependent variables. All statistical analyses were conducted with the free software R©, version 3.6.1.

## Figures and Tables

**Figure 1 plants-11-03536-f001:**
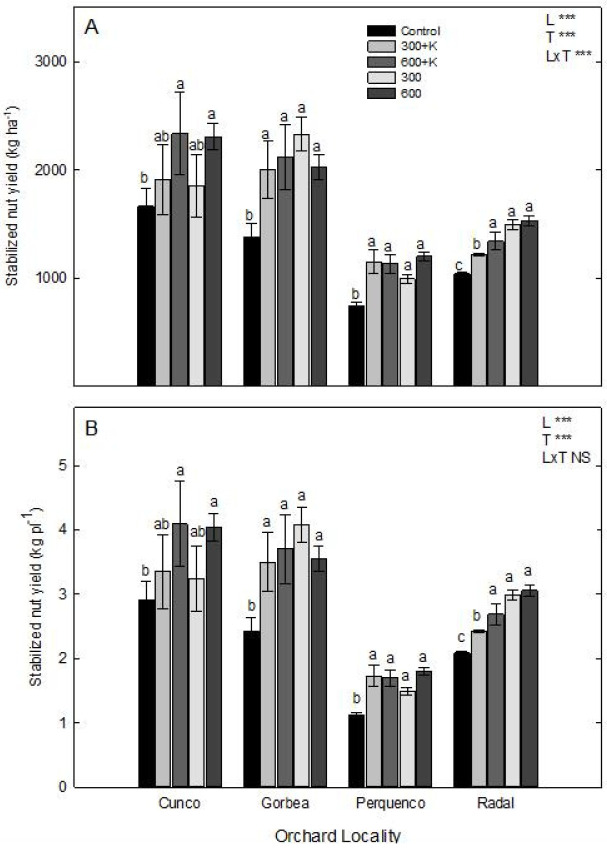
Stabilized nut yield per hectare (kg ha^−1^) (**A**) and per plant (kg pl^−1^) (**B**) in TDG hazelnut after the foliar spraying of Ca, Mg and K in four orchard localities in the La Araucanía region. Bars represent the average of four replicates ± S.E. Different letters indicate statistical differences (*p* ≤ 0.05) between treatments in the same locality. Two-way ANOVA results are shown; NS: not significant; *** *p* < 0.001. Control (water), 300+K (Ca+Mg+K at 300 mg L^−1^), 600+K (Ca+Mg+K at 600 mg L^−1^), 300 (Ca+Mg at 300 mg L^−1^), and 600+K (Ca+Mg at 600 mg L^−1^).

**Figure 2 plants-11-03536-f002:**
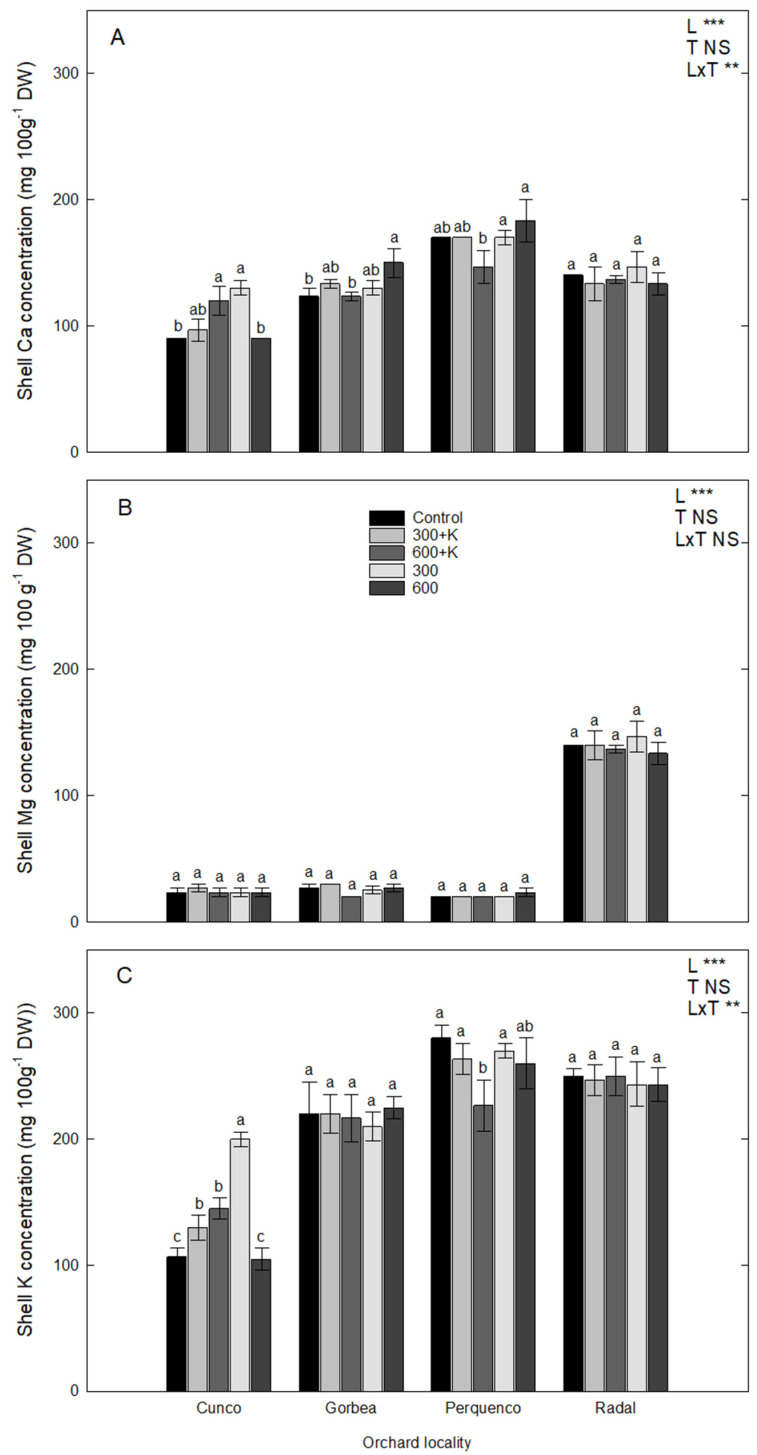
Calcium (**A**), Mg (**B**) and K (**C**) concentrations (mg 100 g^−1^ DW) in the shells of TDG hazelnuts after the foliar spraying of Ca, Mg and K in four orchard localities in the La Araucanía region. Bars represent the average of four replicates ± S.E. Different letters indicate statistical differences (*p* ≤ 0.05) between treatments in the same locality. Two-way ANOVA results are shown; NS: not significant; ** *p* < 0.01; *** *p* < 0.001. Control (water), 300+K (Ca+Mg+K at 300 mg L^−1^), 600+K (Ca+Mg+K at 600 mg L^−1^), 300 (Ca+Mg at 300 mg L^−1^), and 600+K (Ca+Mg at 600 mg L^−1^).

**Figure 3 plants-11-03536-f003:**
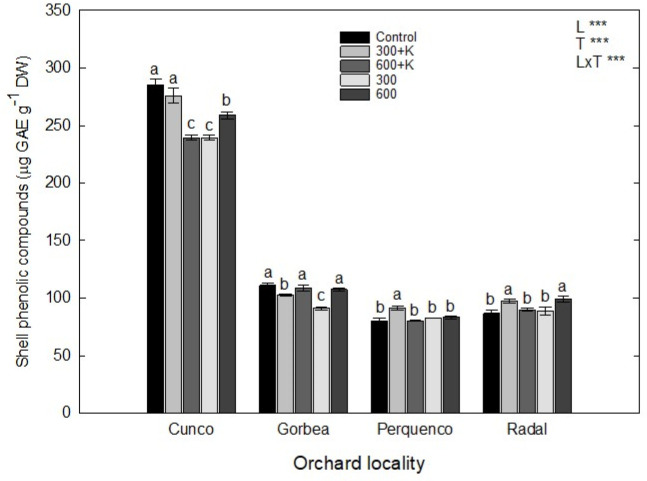
Phenolic compounds (µg GAE g^−1^ DW) in the shells of TDG hazelnuts after the foliar spraying of Ca, Mg and K in four orchards in the La Araucanía region. Bars represent the average of four replicates ± S.E. Different letters indicate statistical differences (*p* ≤ 0.05) between treatments in the same locality. Two-way ANOVA results are shown; *** *p* < 0.001. Control (water), 300+K (Ca+Mg+K at 300 mg L^−1^), 600+K (Ca+Mg+K at 600 mg L^−1^), 300 (Ca+Mg at 300 mg L^−1^), and 600+K (Ca+Mg at 600 mg L^−1^).

**Figure 4 plants-11-03536-f004:**
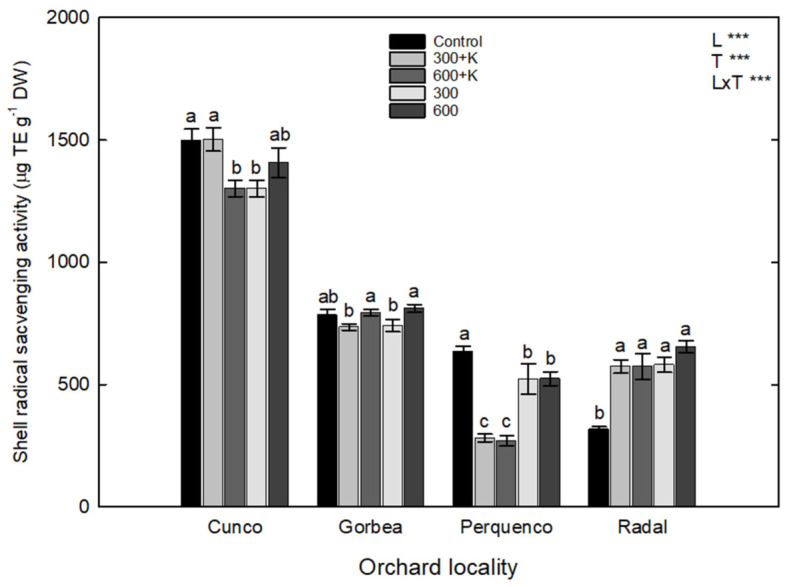
Radical scavenging activity (µg TE g^−1^ DW) of the shells of TDG hazelnuts after the foliar spraying of Ca, Mg and K in four orchards in the La Araucanía region. Bars represent the average of four replicates ± S.E. Different letters indicate statistical differences (*p* ≤ 0.05) between treatments in the same locality. Two-way ANOVA results are shown; *** *p* < 0.001. Control (water), 300+K (Ca+Mg+K at 300 mg L^−1^), 600+K (Ca+Mg+K at 600 mg L^−1^), 300 (Ca+Mg at 300 mg L^−1^), and 600+K (Ca+Mg at 600 mg L^−1^).

**Figure 5 plants-11-03536-f005:**
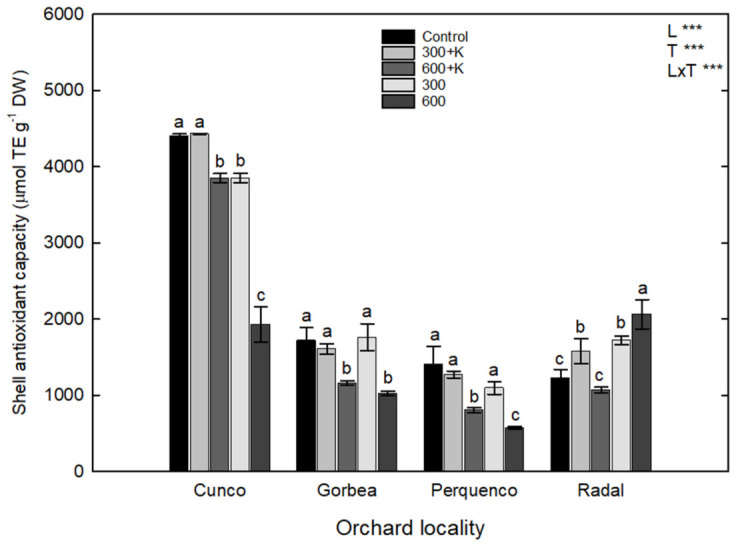
Antioxidant capacity (µmol TE g^−1^ DW) of the shells of TDG hazelnuts after the foliar spraying of Ca, Mg and K in four orchards in the La Araucanía region. Bars represent the average of four replicates ± S.E. Different letters indicate statistical differences (*p* ≤ 0.05) between treatments in the same locality. Two-way ANOVA results are shown; *** *p* < 0.001. Control (water), 300+K (Ca+Mg+K at 300 mg L^−1^), 600+K (Ca+Mg+K at 600 mg L^−1^), 300 (Ca+Mg at 300 mg L^−1^), and 600+K (Ca+Mg at 600 mg L^−1^).

**Figure 6 plants-11-03536-f006:**
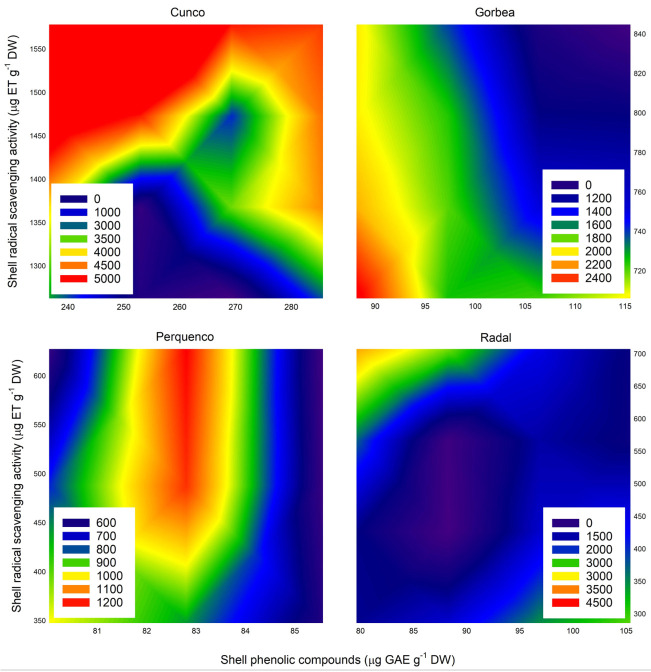
Relationships among shell AC (µmol TE g^−1^ DW), RSA (µg TE g^−1^ DW), and TPC (GAE µg g^−1^ DW) in the shells of TDG hazelnuts after the foliar spraying of Ca, Mg and K in four orchards in the La Araucanía region.

**Table 1 plants-11-03536-t001:** Nut, shell and kernel properties of commercial TDG hazelnuts after the spraying of Ca, Mg and K in four orchards in the La Araucanía region. Values represent the average of four replicates ± S.E. Different low letters indicate statistical differences (*p* ≤ 0.05) between treatments for each locality. Two-way ANOVA results are shown; NS: not significant; ** *p* < 0.01; *** *p* < 0.001. Control (water), 300+K (Ca+Mg+K at 300 mg L^−1^), 600+K (Ca+Mg+K at 600 mg L^−1^), 300 (Ca+Mg at 300 mg L^−1^), and 600+K (Ca+Mg at 600 mg L^−1^).

		Nut			Shell		Kernel
Orchard	Treatment	Weight (g)	Length	Width	Weight (g)	Thickness (mm)	Yield (%)
			(mm)				
Cunco	Control	2.60 ± 0.15 b	17.91 ± 0.27 c	17.86 ± 0.25 a	1.76 ± 0.09 a	1.82 ± 0.20 a	27.94 ± 2.20 b
	300+K	3.17 ± 0.13 a	21.23 ± 0.51 a	17.93 ± 0.55 a	1.83 ± 0.05 a	1.68 ± 0.04 a	42.31 ± 0.70 a
	600+K	2.69 ± 0.18 b	20.38 ± 0.28 b	16.52 ± 0.51 a	1.60 ± 0.09 a	1.57 ± 0.06 a	40.45 ± 1.09 a
	300	2.38 ± 0.14 b	18.97 ± 0.85 c	17.22 ± 0.28 a	1.57 ± 0.03 a	1.72 ± 0.10 a	37.46 ± 0.10 a
	600	2.69 ± 0.21 b	21.23 ± 0.51 b	17.91 ± 0.27 a	1.73 ± 0.13 a	1.69 ± 0.04 a	40.91 ± 0.88 a
	Mean	2.71 ± 0.13	19.94 ± 0.66	17.49 ± 0.28	1.70 ± 0.07	1.70 ± 0.04	37.81 ± 0.34 a
Gorbea	Control	2.56 ± 0.13 a	20.17 ± 0.49 c	17.84 ± 0.33 a	1.50 ± 0.03 a	1.84 ± 0.03 a	41.21 ± 1.70 a
	300+K	2.69 ± 0.81 a	20.36 ± 0.11 a	18.14 ± 0.36 a	1.58 ± 0.49 a	1.99 ± 0.60 a	41.37 ± 1.28 a
	600+K	2.69 ± 0.84 a	20.33 ± 0.30 a	17.78 ± 0.59 a	1.49 ± 0.46 a	1.69 ± 0.51 a	44.38 ± 1.45 a
	300	2.48 ± 0.06 a	20.27 ± 0.14 b	17.38 ± 0.11 a	1.45 ± 0.09 a	1.72 ± 0.06 a	41.83 ± 2.27 a
	600	2.66 ± 0.19 a	20.30 ± 0.06 a	18.07 ± 0.18 a	1.55 ± 0.07 a	1.85 ± 0.09 a	41.27 ± 2.82 a
	Mean	2.66 ± 0.04	20.29 ± 0.03	17.84 ± 0.13	1.51 ± 0.02	1.82 ± 0.05	42.01 ± 0.28
Perquenco	Control	2.53 ± 0.24 a	19.20 ± 0.77 a	17.27 ± 0.62 a	1.39 ± 0.13 a	1.54 ± 0.08 a	44.96 ± 1.67 a
	300+K	2.47 ± 0.12 a	18.71 ± 0.28 a	17.49 ± 0.24 a	1.38 ± 0.01 a	1.75 ± 0.07 a	43.84 ± 2.08 a
	600+K	2.37 ± 0.06 a	19.09 ± 0.15 a	16.92 ± 0.12 a	1.42 ± 0.11 a	1.69 ± 0.13 a	40.28 ± 3.65 a
	300	2.23 ± 0.08 a	18.34 ± 0.38 a	16.69 ± 0.04 a	1.21 ± 0.07 a	1.61 ± 0.10 a	45.53 ± 3.01 a
	600	2.43 ± 0.04 a	18.57 ± 0.19 a	17.53 ± 0.30 a	1.36 ± 0.09 a	1.58 ± 0.07 a	44.19 ± 3.34 a
	Mean	2.41 ± 0.05	18.78 ± 0.16	17.18 ± 0.16	1.35 ± 0.04	1.63 ± 0.04	43.76 ± 0.38
Radal	Control	2.63 ± 0.16 a	18.61 ± 0.23 a	17.48 ± 0.45 a	1.55 ± 0.07 a	1.78 ± 0.05 a	40.98 ± 1.10 a
	300+K	2.39 ± 0.22 a	18.86 ± 0.44 a	16.69 ± 0.57 a	1.39 ± 0.08 a	1.72 ± 0.09 a	41.19 ± 2.34 a
	600+K	2.50 ± 0.22 a	18.99 ± 0.23 a	17.55 ± 0.41 a	1.52 ± 0.10 a	1.72 ± 0.04 a	39.16 ± 1.10 a
	300	2.37 ± 0.04 a	18.61 ± 0.12 a	17.44 ± 0.54 a	1.64 ± 0.09 a	1.78 ± 0.07 a	30.69 ± 4.29 a
	600	2.41 ± 0.11 a	18.96 ± 0.07 a	17.06 ± 0.27 a	1.48 ± 0.10 a	1.73 ± 0.01 a	38.68 ± 3.22 a
	Mean	2.46 ± 0.05	18.81 ± 0.08	17.24 ± 0.16	1.52 ± 0.04	1.75 ± 0.01	38.14 ± 0.62
Significance							
L		**	***	**	***	***	***
T		NS	***	NS	NS	NS	NS
L × T		NS	***	NS	NS	NS	**

**Table 2 plants-11-03536-t002:** Pearson’s correlation coefficients between Ca, Mg, and K concentrations; nut and shell traits; and TPC, RSA and AC in TDG hazelnut shell samples after the spraying of Ca, Mg and K in four orchards in the La Araucanía region. Two-way ANOVA results are shown; * *p* < 0.05; ** *p* < 0.01; *** *p* < 0.001.

	Shell Traits								
	Ca	Mg	K	Weight	Thickness	Kernel Yield	TPC	RSA	AC
Cunco									
Ca	--	0.12	0.79 ***	−0.63 **	−0.18	0.15	0.24	0.28	−0.11
Mg	0.12	--	−0.10	0.54 *	0.17	0.19	0.08	0.23	0.14
K	0.79 ***	−0.10	--	−0.50	−0.06	0.08	0.31	0.35	−0.18
Gorbea									
Ca	--	0.44	0.13	−0.07	0.27	−0.44	−0.42	−0.31	−0.50
Mg	0.44	--	−0.01	−0.28	0.57 *	−0.32	0.15	0.18	−0.35
K	0.13	−0.01	--	0.03	−0.07	0.38	−0.11	−0.07	0.03
Perquenco									
Ca	--	0.47	0.75 **	−0.32	−0.50 *	0.42	0.29	−0.42	−0.03
Mg	0.47	--	0.19	0.31	0.09	−0.37	−0.05	−0.08	0.05
K	0.75 **	0.19	--	−0.27	−0.35	0.35	0.47	−0.45	−0.18
Radal									
Ca	--	0.93 ***	0.75 **	−0.37	−0.30	−0.09	0.22	−0.27	0.25
Mg	0.93 ***	--	0.71 **	−0.44	−0.35	−0.04	0.26	−019	0.23
K	0.75 **	0.71 **	--	−0.78 ***	−0.48	0.08	0.43	0.10	0.01

**Table 3 plants-11-03536-t003:** Spray treatments of Ca, Mg, and K evaluated in the TDG hazelnut orchards of Cunco (38°58′19.8″ S, 72°07′33.0″ W), Gorbea (39°04′23.6″ S, 72°41′48.7″ W), Perquenco (38°25′33.2″ S, 72°21′16.9″ W), and Radal (39°01′06″ S, 72°19′11″ W) in the La Araucanía region during the 2018/19 season.

Treatment (mg L^−1^)			Treatment Code
Ca	Mg	K	
0	0	0	Control
300	300	300	300+K
600	600	600	600+K
300	300	--	300
600	600	--	600

**Table 4 plants-11-03536-t004:** Borehole water samples collected between May and June of 2018.

Chemical Parameter (mg L^−1^)	Cunco	Gorbea	Perquenco	Radal
Ammonia	-	<0.014	<0.014	-
Calcium	-	0.83	14.02	-
Magnesium	-	3.78	6.10	-
Potassium	-	2.34	0.47	-

**Table 5 plants-11-03536-t005:** Chemical parameters in soil samples (0 to 30 cm in depth) from the four studied localities (Cunco, Gorbea, Perquenco, and Radal) were evaluated prior to spray treatment in September 2018. Values represent the mean (*n* = 3).

Chemical Property		Cunco	Gorbea	Perquenco	Radal
N	(mg kg^−1^)	14.33 ± 0.33	15.33 ± 0.66	19.00 ± 0.81	17.66± 0.51
P		5.33 ± 0.33	4.66 ± 0.33	24.5 ± 0.40	4.66 ± 0.33
K		56 ± 10.11	123 ± 20.38	355 ± 19.18	163 ± 13.11
pH (H_2_O)		5.55 ± 0.06	5.58 ± 0.08	5.38 ± 0.07	5.66 ± 0.09
OM	(%)	14.33 ± 0.33	17.66 ± 0.66	14.50 ± 0.40	20.66 ± 0.30
Na	(cmol+/kg)	0.14 ± 0.01	0.12 ± 0.01	0.07 ± 0.01	0.08 ± 0.01
Ca		1.88 ± 0.55	2.38 ± 0.82	6.29 ± 0.80	4.32 ± 0.92
Mg		0.44 ± 0.12	0.65 ± 0.21	0.60 ± 0.07	1.46 ± 0.38
Al		0.14 ± 0.03	0.19 ± 0.06	1.16 ± 0.18	0.11 ± 0.05
CICE		2.68 ± 0.64	3.66 ± 0.98	9.02 ± 0.65	6.38 ± 1.28
*Σ* basis		5.08 ± 0.67	3.47 ± 1.03	7.64 ± 0.84	6.27 ± 1.32
Al sat.	(%)	7.21 ± 3.73	6.98 ± 3.85	13.18 ± 3.04	2.72 ± 2.57

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
