# Peer review of "Antioxidants in Shell and Nut Yield Components after Ca, Mg and K Preharvest Spraying on Hazelnut Plantations in Southern Chile"

_plants, 2022, doi:10.3390/plants11243536_

Round 1

Reviewer 1 Report

The manuscript is interesting and valuable, however, it requires small corrections.

1) Treatment codes should be explained below the figures. At the moment, to understand the figure, one has to follow table 3.

2) Were the nuts washed with water or EDTA after harvesting to remove fertilizer residues? If not, then we are dealing with the determination of elements not in the shell, only in the shell and on its surface.

3) The quality of figure 2 needs to be improved - it is now unreadable.

4) Please correctly spell Latin names, eg Corylus avellana and not Corylus Avellana.

Overall, the work is well written, however, the authors could do much more interesting and valuable analyzes, for example, fat content and composition, amino acid profiles. The manuscript presented by the authors is not very innovative, but it can be valuable from a practical point of view.

Author Response

Dear reviewer

Many thanks for your comments and suggestions. Al changes has been done with track changes as follow:

  1. All treatments codes were added in figure and tables caption for a more easy understanding.
  2. We washed the nuts previously to stabilization, similar to industry. This sentence was included in new version of document.
  3. This figure was improved with a higher resolution of file.
  4. We correct this error.

Thanks again, for yor comments about A new version of MS was uploaded 

Reviewer 2 Report

Your manuscript explores antioxidants in shell and nut yield component after Ca, Mg and K preharvest spraying on hazelnut plantations in southern Chile which has now been carefully reviewed.There are some questions blow:1.I think that if possible, more quarterly surveys will increase the credibility of the results.2.Please replace Figure 2 with the original one. The results shown in Figure 2b show that the effect of LxT seems to be lower than the location effect. Therefore, in the discussion part, it can be discussed why locality of the establishment has a very relevant role.3.Will the variety affect the results?

Thank you!

Author Response

Dear reviewer,

Thank you very much for your comments and suggestions.

In relation to the comments below:

  1. In relation to this comment, we don’t consider more quarterly surveys due to the large harvest and post-harvest periods (we only have one harvest per year) and it would take a long time to consider new quarterly surveys and more data relationed.
  2. The figure 2 is already changed for the original one (improved resolution). In the other hand, the location of plantation is a determining factor, probably due to soil and climatic conditions and their contribution to the synthesis process of the pericarp, its thickness, wide and finally the shell/kernel ratio (Table 4). (paragraph indexed in the document).
  3. We believe that from certain aspects the cultivar can be a factor of variability, especially in terms of biochemistry and concentrations of certain compounds. Moreover, is the most desirable cultivar for exportation due to higher prices paid to growers, during the development of this study only one cultivar (Tonda di Giffoni) was used.